# STIM1–Orai1 Interaction Exacerbates LPS-Induced Inflammation and Endoplasmic Reticulum Stress in Bovine Hepatocytes through Store-Operated Calcium Entry

**DOI:** 10.3390/genes13050874

**Published:** 2022-05-13

**Authors:** Yang Xue, Shendong Zhou, Wan Xie, Meijuan Meng, Nana Ma, Hongzhu Zhang, Yan Wang, Guangjun Chang, Xiangzhen Shen

**Affiliations:** College of Veterinary Medicine, Nanjing Agricultural University, Nanjing 210095, China; 2019107097@njau.edu.cn (Y.X.); 2020107097@stu.njau.edu.cn (S.Z.); 2019207031@njau.edu.cn (W.X.); 2019207045@njau.edu.cn (M.M.); 2017207035@njau.edu.cn (N.M.); 2018107096@njau.edu.cn (H.Z.); 2018207030@njau.edu.cn (Y.W.); changguangjun@njau.edu.cn (G.C.)

**Keywords:** STIM1, Orai1, LPS, inflammation, endoplasmic reticulum stress

## Abstract

(1) Background: The basic mechanism of store-operated Ca^2+^ entry (SOCE) in bovine hepatocytes (BHEC) is related to the activation of STIM1 and Orai1. The effect of STIM1- and Orai1-dependent calcium ion signaling on the NF-κB signaling pathway is unclear. (2) Methods: In this study, the expression of STIM1 and Orai1 in BHEC was regulated. RT-qPCR, Western blotting, and an immunofluorescence antibody (IFA) assay were performed to elucidate the effect of inflammation and endoplasmic reticulum stress (ERS) in BHEC. (3) Results: First of all, in this study, RT-PCR and Western blotting were used to detect the levels of IκB, NF-κB, and inflammatory factors (IL-6, IL-8, and TNF-α) and the expression of genes and proteins related to ERS (PERK, IRE1, ATF6, GRP78, and CHOP), which reached peak levels simultaneously when BHEC were treated with 16 μg/mL LPS for 1 h. For STIM1, we overexpressed STIM1 in BHEC by using plasmid transfection technology. The results showed that after overexpression of STIM1, the gene and protein expression of STIM1 levels were significantly upregulated, and the expression of Orai1 on the cell membrane was also upregulated, which directly activated the SOCE channel and induced inflammation and ERS in BHEC. The overexpression group was then treated with LPS, and it was found that the overexpression of STIM1 could enhance LPS-induced BHEC inflammation and ERS in BHEC. For Orai1, BHEC were pretreated with 8 μg/mL of the specific inhibitor BTP2 for 6 h. It was found that BTP2 could inhibit the expression of mRNA in Orai1, significantly reduce the gene expression of STIM1, inhibit the activation of the NF-κB signaling pathway, and alleviate inflammation and ERS in BHEC under LPS stimulation. (4) Conclusions: In conclusion, STIM1/Orai1 can intervene and exacerbate LPS-induced inflammation and ERS in bovine hepatocytes through SOCE.

## 1. Introduction

The endoplasmic reticulum (ER) is an important organelle in the cells that is involved in various pathways including the folding of newly produced protein, lipid synthesis, and cell signal regulation [1,2]. Endoplasmic reticulum stress (ERS) is an adaptive response that occurs in the ER when the homeostasis of the cell breaks down [3], which induces autophagy, apoptosis, and oxidative stress [4,5], resulting in damage to cells [6]. Previous studies have shown that ERS is associated with stromal interaction molecule 1 (STIM1) and calcium-release-activated calcium channel protein 1 (Orai1) [7,8,9,10,11]. STIM1 is a single-channel-ion-conducting transmembrane protein in the endoplasmic reticulum that participates in the SOCE channel together with Orai1 on the cell membrane [12]. Store-operated Ca^2+^ entry (SOCE) is an external calcium influx induced by a decrease in or depletion of the calcium ion concentration in the ER [13]. The ion channels that mediate SOCE are called store-operated calcium channels (SOCs) [14]. The calcium-release-activated Ca^2+^ channel (CRAC), as one of the most important SOCs, is the main pathway of calcium signal transmission in almost all animal cells. Its function is closely linked to the immune system of the body and it can regulate a variety of immune cells involved in the immune response process [15]. A decrease in the Ca^2+^ concentration in the ER is the initial trigger factor of STIM1 activation [13]. When STIM1 is activated, it will isomerize its conformation and translocate to the cell membrane, and this process is accompanied by the release of Ca^2+^ from the STIM1 domain and the activation of the Orial channel [16]. Normally, Oria1 exists in the form of a dimer and is activated by STIM1 to form a channel protein, which is a highly selective Ca^2+^ channel with its N-terminal forming a hexamer though channel activity [16]. The activation of Orial triggers a small, sustained calcium influx and maintains cellular calcium homeostasis [17].

Lipopolysaccharide (LPS) is a common bacterial endotoxin that is normally produced by bacteria in the gastrointestinal tract and enters blood circulation by absorption [18]. LPS can stimulate the immune system and enhance innate immunity at low concentrations, while it causes extensive and intense inflammatory reactions at high concentrations [19,20]. Some studies in the livers of cows have reported that LPS affects the nutrient metabolism of the liver and reduces the immune capacity of animals, and may even cause endotoxemia in serious cases [21,22]. The liver, as the largest digestive organ and immune organ of dairy cows, is one of the main target organs attacked by LPS and plays an important immune role for LPS [22]. Previous studies have shown that LPS binds to receptors on the cell membrane to activate the monocyte-macrophage system inside the liver [23]. LPS induces the release of TNF, IL-6, IL-8, and other inflammatory mediators in the cell through the mitogen-activated protein kinase pathway, the PKC pathway, the NF-κB pathway, and other inflammatory signaling pathways, resulting in inflammatory reactions [21,24]. It has been reported that upregulation of STIM1/Orai1 can promote inflammation and ERS in mammary tissue [21], and this can inhibit the LPS-induced inflammation by regulating ERS in BHEC [20,25]. LPS stimulation could reduce calcium ions in the ER and activate STIM1/Orai1, and thereby conformational activation, resulting in the influx of Ca^2+^ from the outer membrane and cell damages [21].

Although LPS-induced ERS and inflammation have been reported by many scholars [26,27], the mechanism of the STIM1/Orai1-mediated SOCE channel in LPS-induced inflammation and ERS in bovine hepatocytes remains unclear. In addition, the influence mechanism between STIM1 and Orai1 also needs further research. Therefore, in this article, we sought to understand the actual mechanism and cytoplasmic localization of STIM1–Orai1 interaction with LPS-induced NF-κB signaling pathway and ER stress. Here, our study was mainly focused on controlling the expression of STIM1 and Orai1 on SOCE channels to explore the role of the SOCE channels in LPS-induced liver inflammation and ERS, as well as the underlying mechanisms of SOCE channels in liver inflammation and ERS. 

## 2. Materials and Methods

### 2.1. Chemicals

Lipopolysaccharide (LPS) was purchased from Sigma (0111: B4, Sigma-Aldrich, St. Louis, MI, USA), diluted with PBS (Solarbio Life Sciences, Beijing, China), and reserved. BTP2 was purchased from MCE (YM-58483, MedChemExpress, NJ, USA) and diluted with DMSO (dimethyl sulfoxide, D8371, Solarbio, Beijing, China). Cells were cultured with DMEM-F12, heat-inactivated fetal bovine serum, penicillin, and streptomycin, all purchased from Gibco (Thermo Fisher Scientific, Waltham, MA, USA). STIM1 overexpression plasmid was synthesized by Tsingke (TsingkeBiotechnology, Beijing, China).

### 2.2. Hepatocyte Isolation and Cell Culture

The bovine hepatocytes (BHEC) used in this experiment were kindly provided by Professor Juan J. Loor from the Department of Animal Sciences and Division of Nutritional Sciences at the University of Illinois at Urbana. As in our previous study [22], liver tissues were collected from Holstein cows during the mid-lactation period (about 160 days postpartum) by a non-perfusion technique, and then BHEC were isolated. Cells were cultured in a complete medium containing 10% fetal bovine serum (FBS) and 1% penicillin/streptomycin and incubated at 37 °C and 5% CO_2_ in an incubator. Four to eight generations of cells were used in this study.

### 2.3. Experimental Design

LPS was diluted with double-distilled water to generate a 2 mg/mL stock solution, and BTP2 was diluted to 2, 4, 8, and 12 μg/mL with DMSO. First, the doses of LPS and BTP2 were tested to ensure that the concentration of drugs used in the experiment had no toxic effect on cells. Different time and concentration settings were used explore the best conditions for LPS and BTP2 to induce inflammation and ERS in BHEC. Finally, on the basis of the levels of STIM1/Orai1, inflammatory cytokines, and ERS-related markers, 16 μg/mL of LPS and 8 μg/mL BTP2 were selected for subsequent experiments. Next, to induce the upregulation of STIM1, BHEC were transfected with a STIM1 overexpression plasmid and cultured in a growth medium in 24-well plates for 48 h according to the manufacturer’s instructions. Next, 16 μg/mL LPS was added for 1 h. In addition, after BHECs were cultured in a 24-well plate (4 × 10^4^) for 48 h, the cells were pretreated with 8 μg/mL of the CRAC channel inhibitor BTP2. The cells were collected and analyzed by RT-PCR, Western blotting, and immunofluorescence detection.

### 2.4. Cell Viability

In total, 1 × 10^4^ cells were inoculated in each well of a 96-well plate and cultured for 24 h. The cells were cultured with different concentrations of LPS (0, 5, 10, 15, 20, or 25 μg/mL) or BTP2 (0, 2, 4, 8, 12, or 16 μg/mL) for 12 h, and then 10 μL CCK-8 reagent (Solarbio Life Sciences, Beijing, China) was added to each well and the samples were cultured at 37 °C in an incubator for 3–4 h. The optical density at 450 nm was measured by an enzyme-labeling instrument (Tecan’s Infinite M1000, Thermo Fisher Scientific, Waltham, MA, USA). 

### 2.5. RNA Isolation and Real-Time Polymerase Chain Reaction

The procedures for obtaining the total mRNA of BHEC were performed according to the manufacturer’s instructions. The related chemicals, namely, Trizol reagent (Takara, Dalian, China), isopropanol (Thermo Fisher Scientific, Waltham, MA, USA), trichloromethane (Thermo Fisher Scientific, Waltham, MA, USA), and ethyl alcohol (Thermo Fisher Scientific, Waltham, MA, USA), were used in these steps. The concentration of RNA was measured with a Nano Drop ND-2000 spectrophotometer (Thermo Fisher Scientific, Waltham, MA, USA), and RNA quality was tested by the ratios of A 260 to A 280, although only the level between 1.8 and 2.0 was chosen for later studies. Total RNA (100 ng/µL, diluted with RNase-free water) was reverse-transcribed into complementary DNA (cDNA) with the PrimeScript RT Master Mix kit (catalog no. RR036A; Takara) according to the manufacturer’s instructions, and then the cDNA was diluted fourfold and used for real-time quantitative PCR (RT-PCR). The primer sequences of STIM1/Orai1 and the genes associated with ERS (PERK, IRE1, ATF6, GRP78, CHOP) were designed using sequence data obtained from the National Center for Biotechnology Information with Primer Premier software 5.0 (Premier Bio soft, California, CA, USA), and the primer sequences of the genes for proinflammatory cytokines (IL-6, IL-8, IL-10, IκB, NF-κB, and TNF-α) were directly obtained from previously published articles [21]. The primer sequences of genes were synthesized in Generay equipment (Shanghai, China), and the details are shown in Table 1. Real-time PCR was performed using the SYBR Premix Ex Taq kit (Vazyme, Nanking, China) and the ABI 7300 fast real-time PCR system (Applied Biosystems, Foster City, CA, USA). The changes in the related data were analyzed by the 2^−ΔΔCt^ method, and glyceraldehyde phosphate dehydrogenase (GAPDH) was used as a housekeeping gene to standardize the gene expression.

### 2.6. Western Blot Analysis

As a housekeeping gene, β-actin (1:1000; AA128, Beyotime, Shanghai, China) standardized the difference in protein transfer efficiency. The intensity of target protein bands was quantified by Bio-Rad image system analysis software (Bio-Rad, Hercules, CA, USA).

The cells were inoculated in a 6-well plate with 1 × 10^5^ cells per well. The culture medium was removed after treatment, and the cells were washed 2 times with 1 × PBS; then, the cells were collected by adding a radioimmunoprecipitation assay (RIPA) lysate containing 1% phenylmethylsulfonyl fluoride (PMSF) (Biosharp Life Sciences, Beijing, China). The bicinchoninic acid Protein Assay kit (Thermo Fisher Scientific, Waltham, MA, USA) was used to determine the concentration of protein in the cell lysate, and the protein concentration was diluted to 3 μg/mL in each sample. After adding 5× SDS loading buffer (EpiZyme, Shanghai, China) and denaturing for 5 min at 99 °C, the protein was separated on a 10% or 12.5% SDS-PAGE gel (EpiZyme, Shanghai, China) and then transferred onto a PVDF nitrocellulose membrane. The membrane was blocked with 7% skim milk or BSA (for phosphorylated proteins) at 25 °C for 2 h. The samples were subsequently incubated at 4 °C overnight with protein-specific antibodies (all antibodies were diluted with TBST 1:1000): NF-κB p65 (AF0246, Beyotime), phosphorus-NF-κB P65 (AF5881, Beyotime), IκB α (AF2176, Beyotime), phosphoric acid IκBα (AF1870, Beyotime), TNF α (A11534, Abclonal, Wuhan, China), STIM1 (AF2614, Beyotime), Orai1 (66223-1-Ig, ProteinTech, Thermo Fisher Scientific, Waltham, MA, USA), GRP78 (AF0171, Beyotime), CHOP (AC532, Beyotime), phospho-PERK (AF5902; Cell Signailing, Boston, MA, USA), PERK (D11A8, Cell Signaling), IRE1 α (AI601, Beyotime), phospho-IRE1α (AF5842, Beyotime), and ATF6 (AF6243, Beyotime) at 4 °C overnight. The membrane was incubated with horseradish-peroxidase-coupled secondary antibody (goat anti-rabbit or goat anti-mouse, 1:3000 dilution with TBST) after being fully washed with TBST. As a housekeeping gene, β-actin (1:1000; AA128, Beyotime), was used to standardized for the differences in protein transfer efficiency. The intensity of the target protein bands was quantified by using the Bio-Rad image system analysis software (Bio-Rad, Hercules, CA, USA).

### 2.7. Immunofluorescence

For these assays, 14 mm round cover slides were placed in each well in a 24-well plate for hepatocytes to adhere to the wall. The cells were fixed with 4% paraformaldehyde (300 μL/well, P6148, Sigma-Aldrich) for 15 min, washed with PBS 3 times (5 min each), and perforated with 0.3% Triton X-100 (500 μL/well, T9284, Sigma-Aldrich) for 15 min to improve cell permeability. The cells were washed again with PBS three times, and then the sealant (3% BSA) was added to seal the samples in the incubator at 37 °C for 1 h. Then, primary antibodies p65 (1:100, AF0246, Beyotime), STIM1 (1:200, AF2614, Beyotime), Orai1 (1:200, 66223-1-Ig, ProteinTech), and Calnexin (1:100; AC019, Beyotime) were added, and the mixture was incubated overnight at 4 °C. After the cells were washed with PBS three times, the fluorescent secondary antibodies (1:500, A0562, and A0568, Beyotime) were added and incubated at 37 °C for 1 h without light. After three washes with PBS, the nuclei were stained with a DAPI solution (D8417; Sigma-Aldrich), and a round glass cover was fixed on the glass slide after washing for three times in the dark. The cells were observed by an LSM 710 confocal laser microscope system (Zeiss, Oberkochen, Germany).

### 2.8. Statistical Analysis

All the experiments were repeated more than three times. Data were analyzed by one-way ANOVA with Dunnett’s post hoc test for significance in IBM SPSS 20.0 statistics (IBM Inc., New York, NY, USA). Data were expressed as means and standard errors of the mean. *p* < 0.05 was deemed to be significant, and *p* < 0.01 was deemed to be highly significant. In this experiment, the charts were created by GraphPad Prism 8 software (GraphPad Software, La Jolla, San Diego, CA, USA).

## 3. Results

### 3.1. Viability of BHEC Treated with LPS and BTP2 at Different Doses

The viability of BHECs was detected by a CCK-8 assay after 12 h of treatment with LPS and BTP2 at different concentrations. Compared with the control group (0 μg/mL), LPS at 25 μg/mL reduced the viability of BHEC (*p* < 0.05), but LPS had no cytotoxicity to BHEC in the range of 0–20 μg/mL and could be used in subsequent experiments (Figure 1A). BTP2 treatment at 16 μg/mL significantly decreased cell activity (*p* < 0.05; Figure 1B), but made no difference at 0, 2, 4, 8, and 12 μg/mL compared with the control group and could be used in subsequent experiments.

### 3.2. Optimal Concentrations and Treatment Time Points of LPS

BHEC were treated with 0, 4, 8, 12, 16, or 20 μg/mL LPS for 12 h. The mRNA expression levels of STIM1 and Orai1 were significantly increased in the concentration range of 8 μg/mL to 20 μg/mL, and the highest expression level was found at 16 μg/mL (Figure 2A). We observed the same trend for the protein abundance of STIM1 and Orai1 (Appendix A). NF-κB has long been considered as a typical inflammatory signaling pathway and a key regulator of the immune response, inflammation, and cancer, mainly activated by pro-inflammatory cytokines [28,29]. Inflammatory molecules, such as IL-6, IL-8, and TNF-α, are known to play a pro-inflammatory role in inflammation [29]. Thus, we examined the effects of LPS treatment on BHEC inflammation. As shown in Figure 2B,C, the mRNA expression levels of IκB, NF- κB, IL-6, IL-8, and TNF-α were significantly upregulated at 16 μg/Ml (*p* < 0.01). The protein abundance of PKCα, p65, p-p65, IκB, p-IκB, and TNF-α increased significantly at high concentrations (*p* < 0.01) (Appendix A). Next, the expression of PERK, IRE1, ATF6, GRP78, and CHOP were treated with different doses of LPS (Figure 2D,E). We observed that the protein expression of PERK (*p* < 0.05), IRE1 (*p* < 0.01), ATF6 (*p* < 0.05), GRP78 (*p* < 0.01), and CHOP (*p* < 0.01) increased significantly at 16 μg/mL (Appendix A). Therefore, we chose 16 μg/mL LPS to carry out the subsequent experiments.

Because calcium influx occurred in a short period of time, STIM1 and Orai1 increased significantly after 12 h of LPS treatment. In order to optimize the action time and further reduce the time gradient, BHECs were treated with 16 μg/mL LPS for 0, 1, 2, 3, 4, and 5 h. The expression levels of STIM1 and Orai1 increased significantly at different time points (*p* < 0.01), and neither of the genes increased further with a longer incubation time (Appendix A). At the same time, NF-κB, IκB, IL-6, IL-8, and TNF-α increased to varying degrees during the treatment period (*p* < 0.01; Appendix A), and the ERS-related genes PERK, IRE1, ATF6, GRP78, and CHOP showed a very significant upward trend at each time point (*p* < 0.01; Appendix A). Finally, we chose the most effective treatment time of 1 h for the follow-up experiments.

The immunofluorescence results (shown in Figure 2F) also displayed the same trend. The fluorescence signals of NF-κB p65, STIM1, and Orai1 were observed in the cytoplasm of the control group, and the fluorescence signals were weak. NF-κB p65 is expressed in the nucleus during inflammation and induces the transcription of pro-inflammatory factors [30]. After LPS treatment for 1 h, the immunofluorescence results (shown in Figure 2F) also indicated that the expression of p65 was higher and the location of p65 was predominantly in the nucleus in the LPS group compared with the CON group. Moreover, it was found that the fluorescence signal of STIM1/Orai1 was significantly enhanced, which further indicated that LPS could induce an increase in the expression of STIM1/Orai1. As expected, STIM1/Orai1 played an important role in the regulation of the NF-κB pathway. 

### 3.3. Effects of STIM1 Overexpression on BHEC

Transfection of STIM1 recombinant plasmid significantly enhanced the expression of STIM1 mRNA in cells (Figure 3). Without LPS stimulation, the overexpression of STIM1 (OE-STIM1) increased the mRNA expression of Orai1. Notably, the expression levels of NF-κB, IκB, IL-6, IL-8, and TNFα increased significantly (*p* < 0.01), suggesting that the upregulation of STIM1 is closely related to cellular inflammation. Furthermore, OE-STIM1 significantly increased the mRNA expression of PERK, IRE1, ATF6, GRP78, and CHOP (*p* < 0.01). These results suggested that STIM1 plays a regulatory role in the NF-κB signaling pathway, causing inflammation and ERS in cells. 

### 3.4. Effects of STIM1 Overexpression on LPS-Induced Inflammation and Endoplasmic Reticulum Stress in BHEC

In the previous experiment, we revealed the influence of STIM1 overexpression on BHEC inflammation and ERS. Interestingly, after the overexpression of STIM1, STIM1 and Orai1 increased after treatment 16 μg/mL LPS for 1 h in BHEC, which was significantly different from the results of LPS alone (Figure 4A,D). This indicates that the overexpression of STIM1 can enhance the effect of LPS on the SOCE channel. We also observed that the expression levels of NF-κB (*p* < 0.01), IκB (*p* < 0.01), IL-6 (*p* < 0.01), IL-8 (*p* < 0.01), and TNF-α (*p* < 0.05) in the OE-STIM1 + LPS group were significantly higher than those in the LPS group (Figure 4B). In the OE-STIM1 + LPS group, PKCα (*p* < 0.05), IκB (*p* < 0.01), TNF-α (*p* < 0.01), and p65 (*p* < 0.01) protein expression levels increased significantly (Figure 4E). Although p65 had no significant change, the expression of phosphorylated NF-κB p65 (*p* < 0.01) protein increased significantly. The results of Western blotting suggested that OE-STIM1 exacerbates LPS-induced inflammation in bovine hepatocytes. 

In addition, the mRNA expression and protein abundance of genes related to ERS in BHEC of the OE-STIM1 + LPS group were significantly higher than those of LPS the group (Figure 4C,F). These results suggest that the overexpression of STIM1 could promote ERS after stimulation by LPS in BHEC.

### 3.5. Interaction between STIM1/Orai1 in BHEC and the Effects of STIM1 Overexpression on the NF-κB Signaling Pathway

When STIM1 was overexpressed, the fluorescence signals of STIM1 and Orai1 were significantly enhanced compared with the NC group and were almost the same as those of the LPS group. In the LPS + OE-STIM1 group, STIM1 and Orai1 clustered in the cytoplasm with a significant increase in expression (Figure 5). When Orai1 and STIM1 were stained separately, the staining patterns were different, as shown in Figure 2F. When the two were double-dyed, the staining pattern looked the same, probably because of the interaction between Orai1 and STIM1 [31]. Poor staining of NF-κB p65 and Orai1 in the cytoplasm was observed in the NC group (Appendix A). In contrast, after LPS treatment, the fluorescence signal of NF-κB p65 was transferred into the nucleus and showed strong staining. In the overexpression group, the fluorescence signal of Orai1 increased in extracellular aggregations, and the fluorescence signal of NF-κB p65 was mainly concentrated in the nucleus, which was the same as that of the LPS group. In the LPS + OE-STIM1 group, Orai1 and NF-κB p65 were located in the cytoplasm and nucleus, respectively, and the fluorescence signals were significantly enhanced compared with the other three groups (Appendix A). Calnexin is a calcium-binding protein located on the ER, one that can be combined with a specific oligosaccharide lectin linked to the newly synthesized protein of ER [32]. We used calnexin to locate the ER. The fluorescent signal positions of calnexin and STIM1 proteins overlapped greatly, and the expression of STIM1 was higher after LPS stimulation. In the LPS + OE-STIM1 group, the fluorescence signals of calnexin and STIM1 were enhanced, and their expression levels were increased (Appendix A).

### 3.6. The Best BTP2 Pretreatment Effect

BHEC were pretreated with 0, 2, 4, 8, or 12 μg/mL of BTP2 for 8 h, then the medium was discarded, the cells were washed twice with preheated PBS at 37 °C, and then the cells were treated with 16 μg/mL LPS for 1 h. The results showed that STIM1 and Orai1 were significantly different at the concentration of 8 μg/mL (*p* < 0.05), which significantly reduced the LPS stimulation of the SOCE channel (Appendix A). Therefore, 8 μg/mL BTP2 was selected to optimize the pretreatment time. Then, BHEC were pretreated with 8 μg/mL BTP2 for 0, 1, 2, 3, 4, 5, 6, 7, and 8 h, and the expression levels of STIM1 and Orai1 were detected. We observed that LPS could not significantly increase Orai1 after treatment with BTP2, the expression levels of Orai1 were significantly different at 6 h and 7 h, and STIM1 had also significantly decreased at 6 h (Appendix A). Therefore, after comprehensive consideration, we chose the treatment time of 6 h for the follow-up test.

### 3.7. Effects of the Orai1 Inhibitor BTP2 on SOCE

As shown in Figure 6A, the pretreatment group with BTP2 alone was unable to induce the activation of the SOCE channel and had no significant effect on BHEC, and the mRNA expressions levels of STIM1 and Orai1 showed no significant difference compared with the control group. However, LPS stimulation of BTP2-treated BHEC significantly reduced the expression of STIM1 and Orai1, even though it had no significant difference from the control level, indicating that the Orai1 inhibitor BTP2 can effectively reduce the activation of Orai1 and block the activation of the SOCE channel.

### 3.8. The Orai1 Inhibitor BTP2 Alleviated LPS-Induced BHEC Inflammation and ERS

Compared with the LPS group, the expression levels of NF-κB and IκB, and the expression levels of the inflammatory factors IL-6, IL-8, and TNF-α decreased after pretreatment with BTP2 and stimulation by LPS (*p* < 0.01; Figure 6B,C). The mRNA expression levels of the ERS-related genes PERK, IRE1, ATF6, GRP78, and CHOP also decreased significantly (*p* < 0.01; Figure 6D). The protein expression levels were consistent with the mRNA expression trends (Figure 6F,G). This suggests that after BTP2 suppresses the expression of Orai1, the SOCE channels are silenced, which effectively alleviates inflammation and the ERS effects of LPS on BHEC. When Orai1 was inhibited, LPS stimulation did not enhance the fluorescence signals of STIM1 and Orai1, while p65 nucleation decreased and fluorescence intensity weakened (Figure 7). LPS stimulation alone enhanced the fluorescence intensity of STIM1 in the cytoplasm and the expression of calnexin protein in the ER (Appendix A). After pretreatment with BTP2 and LPS stimulation, the fluorescence intensity returned to the control group’s level, and the protein positions of calnexin and STIM1 overlapped greatly (Appendix A).

## 4. Discussion

It is well known that LPS-induced inflammation and endoplasmic reticulum stress can lead to liver injury, which has been extensively studied [23,28,33]. In our research, we found that 16 μg/mL LPS could better induce inflammation and ERS in BHEC. In contrast, the expression of ERS-related markers decreased under the stimulation with 20 μg/mL LPS, which may have been related to autophagy in the hepatocytes [34]. ERS has previously been shown to be a potent trigger for inducing autophagy, and acute liver injury may lead to ERS, which subsequently activates autophagy [33]. Autophagy is known to modulate normal liver function [35,36,37], which may be why ERS markers are lower at 20 μg/mL than at 16 μg/mL. During selection of the optimal concentration of LPS, STIM1 mRNA levels increased from 4 μg/mL LPS and above, whereas a similar increase was not observed for STIM1 protein. LPS, treatment at low concentrations can only induce expression at the gene level but cannot increase protein expression, which requires transcriptional modification. There was a similar result in Cheng’s study [24].

Existing studies have shown that inhibition of the NF-κB signaling pathway can reduce LPS-induced liver dysfunction and inflammation [23]. Inflammatory responses are characterized by the co-activation of various signaling pathways that promote the expression of pro-inflammatory and anti-inflammatory mediators in tissue cells [38]. Orai1 is known to be involved in the NF-κB signaling pathway [39]. Therefore, the role of STIM1/Orai1 in NF-κB inflammatory signaling needs further study. In this study, we verified that LPS can indeed upregulate the expression of STIM1 and Orai1 in BHECs and promote the abundance of pro-inflammatory factors and key factors in ERS. These data are consistent with the results of Meng et al. [21]. This demonstrated that LPS can activate the STIM1/Orai1-mediated SOCE channel and induce an inflammatory response and ER stress in bovine hepatocytes. Nevertheless, there are gaps in the knowledge regarding the cell-specific mechanisms of STIM1- and Orai1-mediated inflammation and ERS. The actual mechanism and cytoplasmic localization of STIM1/Orai1 interaction with LPS-induced NF-κB signaling pathway and ERS are poorly understood.

Studies have shown that physiological responses such as inflammation and ERS may be regulated by SOCE, which is a very complex physiological process in the body [40,41]. Continuous ER stress is considered to be a pathogenic factor of many chronic diseases because it may trigger abnormal inflammatory signals and promote cell death [42]. In addition, ER stress responses may contribute to disease progression by coordinating destructive inflammatory responses [8]. Therefore, we conducted targeted studies on the interaction between STIM1 and Orai1 in the SOCE pathway. In this research, we used the STIM1 plasmid transfection technique to overexpress STIM1 gene in hepatocytes in order to study the effects of increased expression of the STIM1 gene on the SOCE channel, liver inflammation, and ER stress. Our results showed that the upregulation of STIM1 in bovine hepatocytes could directly affect the expression levels of Orai1. The activation of Orai1 by STIM1 was also described in the review by Lunz et al. [16]. It is well known that the most direct response to ERS in hepatocytes is the dissociation of the ER transmembrane receptors IRE1α (inositol requiring enzyme 1α), PERK (pancreatic ER kinase), and ATF6 (activating transcription factor 6), as well as glucose regulatory protein 78 (GRP78)/Bip, triggering the ER self-protection mechanism [8,10]. Once ER stress reaches a threshold, apoptosis induced by C/EBP homologous protein (CHOP) occurs [36]. In our study, the transcription levels of PERK, IRE1, and ATF6 were significantly elevated, indicating that the ER received signals from the SOCE channel, activating PERK, IRE1, and ATF6; upregulating GRP78 and CHOP expression levels; and initiating the ER stress signaling mechanism. In addition, the key transcription factors NF-κB and IκB in the NF-κB signaling pathway were also activated by the increased expression of STIM1, thereby promoting the expression of the inflammatory factors IL-6, IL-8, and TNF-α in our results. Next, BHEC after overexpression of STIM1 was exposed to LPS, the results showed that overexpression of the STIM1 could enhance the effect of LPS, and the inflammatory effect and the ERS level were significantly increased compared with those in BHEC stimulated by LPS alone. The immunofluorescence results showed that after upregulation, STIM1 and Orai1 clustered in points in the cytoplasm, and their expression levels increased significantly, indicating a co-localization phenomenon. This means that the fluorescence signal of STIM1 was correlated with that of Orai1. When Orai1 and STIM1 were stained separately, the staining patterns were different, as shown in Figure 2F. When the two are double-dyed, the staining pattern looks the same, probably because of the interaction between Orai1 and STIM1 [31]. Therefore, we concluded that STIM1, as a calcium ion receptor, plays an important role in the activation of the SOCE channel, which can activate Orai1 on the membrane, causing inflammation and ERS, which contribute to the LPS-toxic environment. Thus, STIM1 acts as a gateway to the SOCE channel, playing an irreplaceable role in SOCE, accepting calcium signaling to induce ERS, and turning on NF-κB signaling to induce inflammation in hepatocytes.

Zhang et al. found that the increased expression of Orai1 can cause mitochondrial dysfunction and oxidative stress in hepatocytes [11]. When the ER’s calcium concentration is reduced, Orai1 protein is activated and promotes calcium influx [11,40]. In Waldron’s study, the Orai1 inhibitor CM4620 was found to alleviate acute pancreatitis in rats, decreasing neutrophil oxidation bursts and inflammatory gene expression during pancreatitis [43]. There is ample evidence that SOCE, via Orai1, promotes the expression of inflammation genes in animals [44]. However, the role of Orai1 in LPS-induced hepatocyte inflammation and ERS remains unclear. Therefore, in order to study the role of Orai1 and its mediation in the SOCE channel, we specifically inhibited the expression of Orai1 in hepatocytes. BTP2 is an inhibitor of the Ca^2+^ channel Orai1, which is widely used in skeletal muscle [45]. Inhibition of SOCE channel with BTP2 prior to LPS treatment resulted in the opposite result to overexpression of STIM1 (Figure 6). Transcriptional levels of STIM1 and Orai1 decreased successfully, while LPS lost its effect under the inhibition by BTP2 in BHEC. With the closing of the SOCE channel, the expression levels of the inflammatory factors IL-6, IL-8, and TNF-α in the hepatocytes showed low levels, and the expression levels of the genes and proteins of the endoplasmic reticulum stress markers p-PERK, PERK, p-IRE1, IRE1, ATF6, GRP78, and CHOP were consistent with this. We think that Orai1 may contribute to endoplasmic reticulum stress because the silence of Orai1 prevents the abundance of ER stress marker GRP78 and ER stress sensors PERK, IRE1, ATF6, and CHOP from increasing. In addition, the protein fluorescence signals of STIM1/Orai1, p65/Orai1, and calnexin/STIM1 in hepatocytes were collected (Figure 7). The results verified the fluorescence results of overexpression of STIM1, and BTP2 was able to reduce the overlapping of STIM1/Orai1 by LPS and inhibit the SOCE channel. At the same time, it prevented NF-κB p65 from entering the nucleus and activating, thus hindering the activation of the NF-κB signaling pathway, reducing the inflammatory reaction, and relieving endoplasmic reticulum stress. Therefore, we speculated that STIM1 and Orai1 are like the front and back doors of SOCE, controlling the opening and closing of the SOCE channel. 

## 5. Conclusions

In summary, on the basis of the existing experimental data, we concluded that STIM1 and Orai1 have a mutual intervention mechanism, and their mediation of SOCE can activate the NF-κB signaling pathway, regulate the immune response, and promote inflammation and ER stress in hepatocytes. Through its activation and inactivation mechanism, the STIM1/Orai1 protein not only participates in regulating the opening and closing of the SOCE pathway, but also participates in the regulation of important intracellular functional activities (Figure 8). Therefore, the STIM1/Orai1 protein is expected to be a potential new target and biomarker for the treatment of a variety of diseases.

## Figures and Tables

**Figure 1 genes-13-00874-f001:**
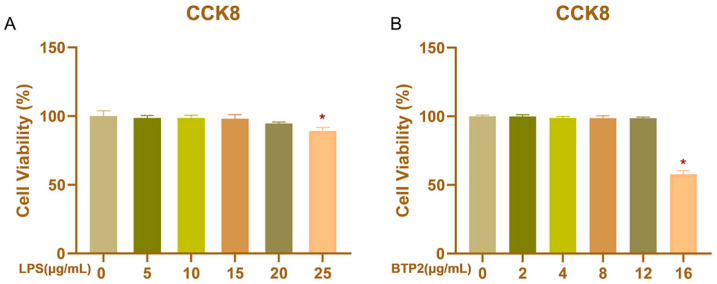
The effect of LPS and BTP2 on the viability of bovine hepatocytes (BHECs) was detected by a CCK-8 assay. BHECs were exposed to 0, 5, 10, 15, 20, or 25 μg/mL LPS for 12 h (**A**) or were treated with 0, 2, 4, 8, 12, or 16 μg/mL BTP2 for 12 h (**B**). Absorbance was measured at 450 nm to evaluate cell viability. Error bars represent the mean ± SEM (*n* = 6). * *p* < 0.05 compared with the controls (0 μg/mL).

**Figure 2 genes-13-00874-f002:**
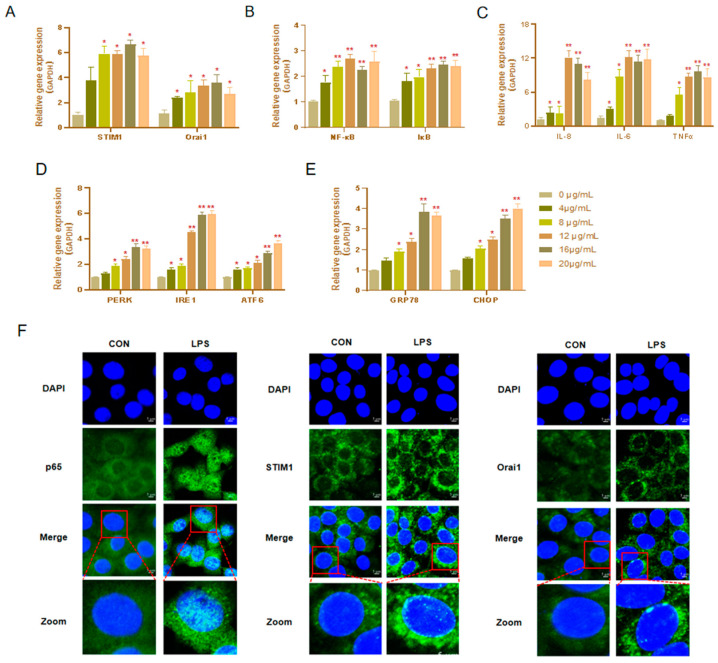
Effects of different concentrations of LPS on bovine hepatocytes (BHEC). When the adherent density of BHEC reached about 70%, the BHEC was treated with LPS at 0, 4, 8, 12, 16, or 20 μg/mL for 12 h. Gene expression was normalized with GAPDH as the reference protein. (**A**) Gene expression of STIM1 and Orai1 in BHECs. (**B**,**C**) Gene expression levels of NF-κB, IκB, IL-6, IL-8, and TNF-α. (**D**,**E**) Changes in endoplasmic reticulum stress (ERS)-related genes (PERK, IRE1, ATF6, GRP78, CHOP) under different doses of LPS. Error bars represent the means ± SEM (*n* = 3, *n* = 3). * *p* < 0.05, ** *p* < 0.01 compared with the controls. (**F**) After treatment with 16 μg/mL LPS for 1 h, the fluorescence signals and position changes of STIM1, Orai1, and p65 in BHEC were observed by confocal immunofluorescence microscopy. DAPI blue fluorescence was used to label the location of the nucleus; FITC green fluorescence was used to label the target proteins.

**Figure 3 genes-13-00874-f003:**
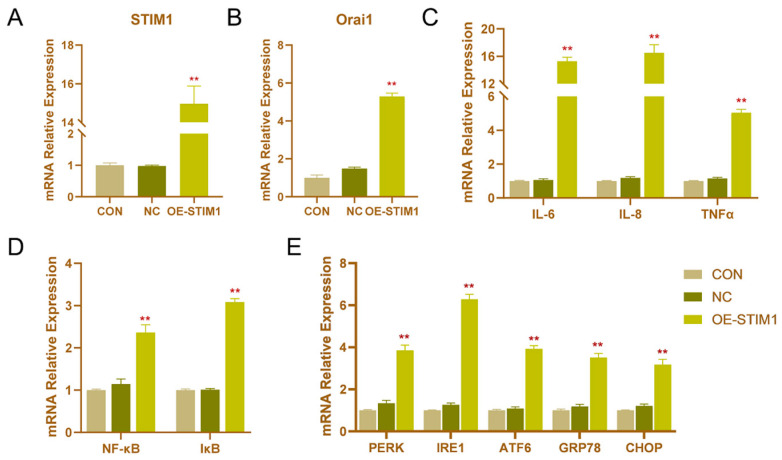
Effects of the overexpression of STIM1 on inflammation and endoplasmic reticulum stress (ERS) in BHEC. Treatments: BHEC were transfected with overexpressed plasmids and cultured for 48 h. CON, complete control groups; NC, transfected blank plasmid and cultured for 48 h; OE-STIM1, BHEC transfected with overexpressed STIM1 plasmid cultured for 48 h. All groups were cultured without LPS. (A,B) Expression of (**A**) STIM1 and (**B**) Orai1 in BHECs. (**C**,**D**) Expression of (**C**) IL-6, IL-8, and TNF-α, and (**D**) NF-κB and IκB associated with the inflammatory response. (**E**) Expression levels of ERS-related genes (PERK, IRE1, ATF6, GRP78, and CHOP) in BHEC. Gene expression was normalized to the expression of GAPDH. The results are presented as the means ± SEM., ** *p* < 0.01.

**Figure 4 genes-13-00874-f004:**
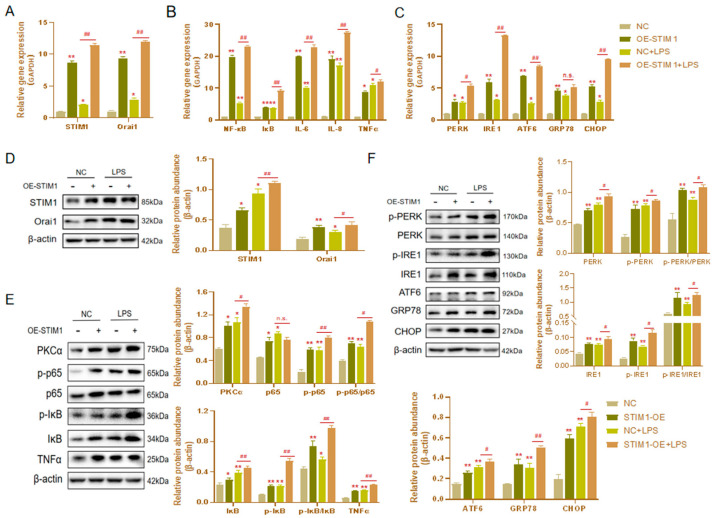
Effects of STIM1 overexpression on LPS-induced inflammation and endoplasmic reticulum stress (ERS) in BHEC. Treatments: BHEC were transfected with overexpressed plasmids and cultured for 48 h. NC, transfected blank plasmid cultured for 48 h without LPS treatment; OE-STIM1, BHEC transfected with overexpressed STIM1 plasmids that were cultured for 48 h without LPS treatment; LPS, transfected blank plasmids cultured for 48 h, then treated with 16 μg/mL LPS for 1 h; OE-STIM1 + LPS, BHEC transfected with overexpressed STIM1 plasmids that were cultured for 48 h and then treated with 16 μg/mL LPS for 1 h. Gene expression of STIM1/Orai1 (**A**), inflammatory cytokines (NF-κB, IκB, IL-6, IL-8, and TNF-α) (**B**), and ERS-related genes (PERK, IRE1, ATF6, GRP78, and CHOP) (**C**) in BHEC. The protein abundance of STIM1/Orai1 (**D**), inflammatory cytokines (PKCα, p65, p-p65, IκB, p-IκB, and TNFα) (**E**), and ERS-related genes (PERK, IRE1, ATF6, GRP78, and CHOP) (**F**) was normalized to the abundance of β-actin. The results are presented as the means ± SEM. * *p* < 0.05, ** *p* < 0.01 indicate that the LPS groups and the OE-STIM1 groups were significantly different from control group. ^#^
*p* < 0.05, ^##^
*p* < 0.01 indicates the significant difference between the OE-STIM1 + LPS groups and the NC + LPS groups; n.s. indicates no difference between the two groups.

**Figure 5 genes-13-00874-f005:**
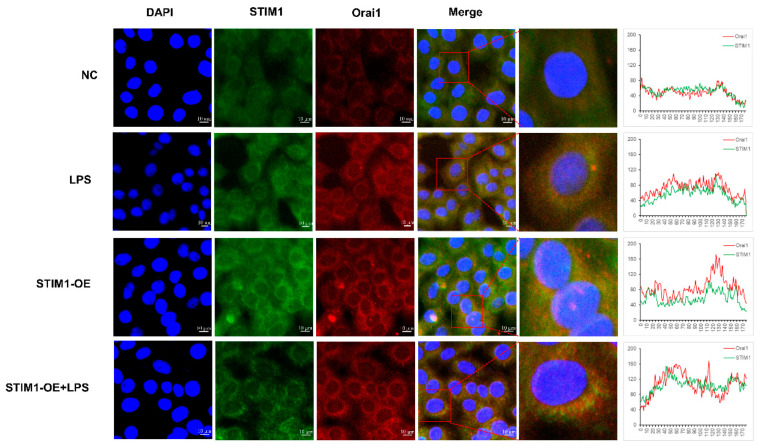
Protein expression and location of STIM1/Orai1 in BMEC. DAPI blue fluorescence was used to label the nuclear locations, FITC green fluorescence was used to label STIM1 protein, and Cy3 red fluorescence was used to label Orai1 protein. ImageJ was used to draw the line graph to show the fluorescence co-location trends.

**Figure 6 genes-13-00874-f006:**
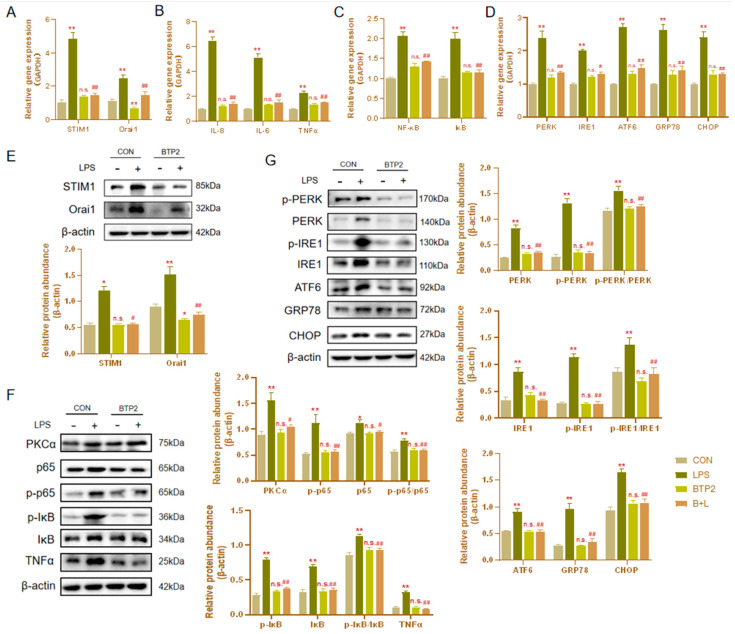
Effects of the Orai1 inhibitor BTP2 on LPS-induced inflammation and endoplasmic reticulum stress (ERS) in BHEC. Treatments: BHEC were cultured in a BTP2-free medium for 24 h. CON, complete control groups; LPS, cells treated with 16 μg/mL LPS for 1 h; BTP2, pretreatment with 8 μg/mL BTP2 for 6 h without LPS treatment; LPS + BTP2, 8 μg/mL BTP2 for 6 h, followed by 16 μg/mL LPS for 1 h. (**A**–**D**) Expression levels of STIM1/Orai1 and the genes related to inflammatory cytokines (NF-κB, IκB, IL-6, IL-8, and TNFα) and ERS (PERK, IRE1, ATF6, GRP78, and CHOP) were shown. Gene expression levels were normalized to that of GAPDH. (**E**–**G**) Protein abundance levels of STIM1/Orai1, PKCα, p65, p-p65, IκB, p-IκB, TNFα, PERK, p-PERK, IRE1, p-IRE1, ATF6, GRP78, and CHOP in BHEC, showing representative bands of the Western blot analysis and the quantified volume of specific bands. The results are presented as the means ± SEM. * *p* < 0.05, ** *p* < 0.01 indicate a significant difference between the LPS groups and the control groups; n.s. indicates that there was no significant difference between the BTP2 group and the CON group. ^#^
*p* < 0.05, ^##^
*p* < 0.01 indicate a significant difference between the BTP2 pretreatment groups and the LPS groups.

**Figure 7 genes-13-00874-f007:**
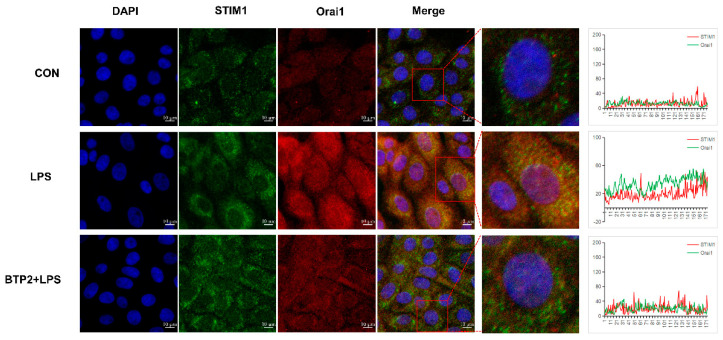
Protein expression and location of STIM1/Orai1 in BMEC. DAPI blue fluorescence was used to label the nuclear location, FITC green fluorescence was used to label STIM1 protein, and Cy3 red fluorescence was used to label Orai1 protein. ImageJ was used to draw the line graph to show the fluorescence co-location trends.

**Figure 8 genes-13-00874-f008:**
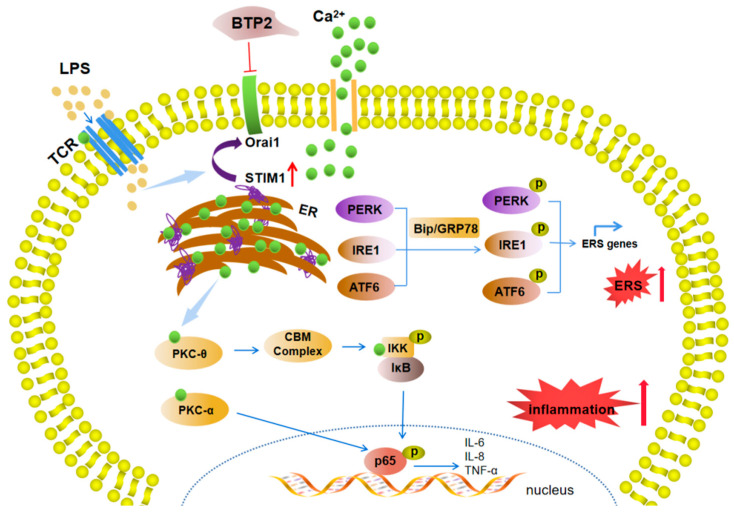
Schematic representation shows that Toll-like receptors are stimulated by LPS, activate the STIM1/Orai1-mediated SOCE channel, cause endoplasmic reticulum stress (ERS), and turn on the NF-κB signaling pathway to induce inflammation. The upregulation of STIM1 intensifies LPS-induced inflammation and endoplasmic reticulum stress (ERS), whereas the inhibition of Orai1 alleviates the inflammatory effects of LPS and ERS. LPS, lipopolysaccharide; STIM1, stromal interaction molecule 1; Orai1, calcium release-activated calcium channel protein 1; NF-κB, nuclear factor-kappa B; IκB, inhibitor of NF-κB; PERK, pancreatic elf-2 kinase pancreatic ER kinase; ATF6, activating transcription factor 6; IRE1, inositol requiring enzyme 1; GRP78, glucose-regulated protein 78; CHOP, C/EBP homologous protein; IL-6, interleukin-6; IL-8, interleukin-8; TNF-α, tumor necrosis factor-α.

**Table 1 genes-13-00874-t001:** The list of primer sequences for RT-qPCR.

Gene	Gene Bank Accession		Primer Sequence (5′-3′)	Length (bp)
*STIM1*	NM_001035409.1	Forward	AGTCCACTGCAGCAGAGTTT	250
		Reverse	TTCCATGCCTTCCACAGGTC	
*Orai1*	NM_001099002.1	Forward	GTCCTGGCGCAAACTCTACT	125
		Reverse	GGTAGTCGTGGTCAGCATCC	
*IL-6*	NM_173923.2	Forward	CGAAGCTCTCATTAAGCACATC	221
		Reverse	CCAGGTATATCTGATACTCCAG	
*IL-8*	NM_173925.2	Forward	CTGAGAGTTATTGAGAGTGGGC	259
		Reverse	CAGTACTCAAGGCACTGAAGTAG	
*TNF-α*	NM_173966.3	Forward	CAACAGGCCTCTGGTTCAGAC	209
		Reverse	GGACCTGCGAGTAGATGAGG	
*IL-10*	NM_174088.1	Forward	AGCACTACTCTGTTGCCTGG	230
		Reverse	TTGGGGTAGACTTTGGGGTCT	
*NF-κB*	NM_001045868.1	Forward	CTTCCATCCTGGAACCACTAAA	108
		Reverse	ACCTCTCTGTCGTCACTCTT	
*IκB*	XM_0056999961	Forward	GGTGAAGGAGCTGCGAGAG	326
		Reverse	GCTCACAGGCAAGGTGTAGG	
*PERK*	NM_001098086.1	Forward	GCCGCTCAGCTCTCCTAGTCC	165
		Reverse	TGGCTCTCGGATGAACTGGTCTG	
*IRE1*	XM_024980954.1	Forward	TCCTCCCAGATCCCAACGAT	127
		Reverse	ATGCCATCTGAACTTCGGCA	
*ATF6*	XM_024989876.1	Forward	AGCCCTGATGGTGCTAACTGA	100
		Reverse	TTCATGATTTAACCTGAGAGATTCTGTT	
*GRP78*	XM_024998380.1	Forward	AACGACCCCTGACGAAAGAC	100
		Reverse	TCAAAGGTGACTTCAATCTGTGG	
*CHOP*	XM_019960966.1	Forward	CCTGAGGAGAGAGTGTTCCAG	160
		Reverse	CCTGCAGGTCCTCATACCAG	
*GAPDH*	NM_001034034.2	Forward	GGGTCATCATCTCTGCACCT	180
		Reverse	GGTCATAAGTCCCTCCACGA	

## Data Availability

The datasets generated and/or analysed during the current study are available in the article and the Appendix A.

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
