# Peer review of "STIM1–Orai1 Interaction Exacerbates LPS-Induced Inflammation and Endoplasmic Reticulum Stress in Bovine Hepatocytes through Store-Operated Calcium Entry"

_genes, 2022, doi:10.3390/genes13050874_

Round 1
Reviewer 1 Report
The role of STIM1 and ORAI1 in LPS-induced ER stress response and NF-KB signaling is an important knowledge gap. The primary finding of this study is that LPS treatment induces STIM1 and ORAI1 expression in primary hepatocytes and is associated with activation of ER stress and NF-KB signaling pathway. There are several significant concerns regarding the interpretation of the experimental results.
MAJOR:
- Figure 1: Why is a dose dependent response on cell viability not observed for treatment with LPS and BTP2? For instance, the significant decrease in cell viability at 16 ug/ml for BTP2 treatment is unexpected compared to 12 ug/ml
- Figure 2F: Which of the two bands corresponds to ORAI1? How was antibody specificity determined?
- Figure 2A and 2F: Authors should discuss why STIM1 mRNA levels increase from 4 ug/ml LPS treatment onwards, whereas a similar increase is not observed for STIM1 protein
- Figure 2G: Phosphorylated protein levels should be normalized to the respective total protein
- Figure 2G: No TNFa band is visible in the representative blot at 20 ug/ml treatment; however, quantitation of the data is shown.
- Figure 2F: Authors should discuss why several ER stress markers decrease at 20 ug/ml treatment compared to 16 ug/ml treatment.
- Phosphorylation of PERK, IRE1a and cleavage of ATF6 represent ER stress activation. Western blotting should be performed for the respective proteins to confirm ER stress
- In legend for figure 2, please specify that gene expression and protein abundance were normalized to GAPDH and actin rather than corrected.
- Figure 3F: Representative image for p65 does not demonstrate increased expression after LPS treatment
- Authors refer to mRNA abundance measured by qRT-PCR as ‘secretion levels’ throughout the manuscript. mRNA abundance measured by qPCR does not indicate secretion levels
- Authors refer to FITC ‘blue’ fluorescence. Please correct to ‘green’ fluorescence.
- Figure 5D: Dramatic increase in ORAI1 protein is not seen concurrent with mRNA increase
- Figure 5G has several issues:
- ORAI1 and STIM1 staining pattern look identical
- Nuclear expression of STIM1 and ORAI1 is observed in STIM1-OE panels
- ORAI1 localization and expression patterns are different in panels G & H under STIM1-OE and STIM1-OE+LPS conditions.
- Figure 5I: Diffuse pattern for calnexin is unexpected as it is an ER localized protein
- Authors should discuss why STIM1-OE and LPS treatment increase CHOP expression, however, cell viability is unaffected (in LPS treatment).
- Line 323: “STIM1 was hypopolymerized” observation is not supported by data
- Line 328: “preheated PBS” was PBS heated to a specific temperature?
- Line 361: “BTP2 suppresses the expression of Orai1” data in figure 7A does not support suppression of Orai1 expression by BTP2
- Figure 7F: p65 representative blot does not correspond with the quantitation
MINOR:
- Language needs to be edited significantly to improve clarity
- Please provide source of all chemicals used in the study.
- Section 2.2:
- “As in our previous study…” please provide reference for study.
- “which was treated…” please specify treatment conditions in brief.
- Section 2.3: What is the source of the STIM1 overexpression plasmid?
- Section 2.4: please specify the name of the enzyme labeling instrument.
- Section 2.5 and 2.6 require significant editing for clarity.
- Authors should consider moving figures corresponding to dose and timing to supplemental data
Reviewer 2 Report
The manuscript by Xue et al. reports that STIM1/Orai1 Interaction exacerbates LPS-induced inflammation and endoplasmic reticulum stress in bovine hepatocytes through Store-Operated Calcium entry. Thus the manuscript contains a set of novel data. However, I have serious concerns:
- First of all I have to point the attention on the fact that English is very poor. At some parts the logic is missing and poor English is the cause:
- The study represented in the manuscript is poorly connected with already published works on the topic (https://pubmed.ncbi.nlm.nih.gov/?term=STIM1%20and%20Orai1%20and%20lps&sort=pubdate). The most important papers are even not cited. The most important question: what novel findings (except the model system) in comparison to published works are reported by the authors? Thus in my opinion, prior to the repeated review process, the manuscript should be reorganized drastically: the Introduction should be added more information on the published works and contain clear objectives of the study at the last paragraph, while the Discussion should clearly state what novel findings this study brings to the general knowledge. Results section should be reorganized as well following the objectives.
Round 2
Reviewer 1 Report
The authors have addressed all of my comments and incorporated all of the necessary changes. Authors are encouraged to provide insets for figure 2F focusing to an individual cell, particularly for p65, that highlights the nuclear translocation.
Author Response
Thank you for your comments and suggestions. A more representative image of p65 has been selected in Figure. 2F, and a small diagram of an individual cell has been cut out to show it.
Reviewer 2 Report
I greatly appreciate the authors for their efforts.
Author Response
Thank you for your comments and suggestions.